# Outcomes of a Standardized, High-Caloric, Inpatient Re-Alimentation Treatment Protocol in 120 Severely Malnourished Adolescents with Anorexia Nervosa

**DOI:** 10.3390/jcm11092585

**Published:** 2022-05-05

**Authors:** Sophia Dalenbrook, Silke Naab, Andrea K. Garber, Christoph U. Correll, Ulrich Voderholzer, Verena Haas

**Affiliations:** 1Department of Child and Adolescent Psychiatry, Charité—Universitätsmedizin Berlin, Corporate Member of Freie Universität Berlin and Humboldt-Universität zu Berlin, Augustenburger Platz 3, 13353 Berlin, Germany; christoph.correll@charite.de; 2Schoen Clinic Roseneck, Prien am Chiemsee, 83209 Prien am Chiemsee, Germany; snaab@schoen-klinik.de (S.N.); uvoderholzer@schoen-klinik.de (U.V.); 3Division of Adolescent and Young Adult Medicine, Department of Pediatrics, University of California, San Francisco, CA 94143, USA; andrea.garber@ucsf.edu; 4Donald and Barbara Zucker School of Medicine at Hofstra/Northwell, Hempstead, NY 11549, USA; 5Department of Psychiatry, The Zucker Hillside Hospital, Glen Oaks, NY 11004, USA; 6Department of Psychiatry and Psychotherapy, University Hospital, Ludwig Maximillians Universität München, 80539 Munich, Germany; 7Department of Psychiatry and Psychotherapy, University Hospital of Freiburg, 79106 Freiburg, Germany

**Keywords:** energy intake, eating disorders, refeeding syndrome, nutrition

## Abstract

Evidence accumulates that, with close medical monitoring and phosphate supplementation, higher-caloric re-alimentation protocols beginning at 2000 kcal/day (HCR) are not associated with an increased incidence of electrolyte abnormalities in patients with anorexia nervosa (AN) but rather result in faster weight gain. These studies are still scant and have largely been performed in adults or moderately malnourished adolescents. Methods: A retrospective chart review of patients with AN aged 12–20 years and with a body mass index (BMI) < 15 kg/m^2^ alimented according to a standardized treatment protocol in a German clinic specialized in AN was conducted. All patients received 2000 kcal/day from day one. The effect of HCR was examined with respect to laboratory changes and weight development over 4 weeks. Results: In 120 youth (119 (99.2%) females and 1 (0.8%) male, the mean BMI was 13.1 ± 1.1 (range = 10.2–15.0), %mBMI was 62.1 ± 6.0% and weight gain was 0.76 ± 0.22 kg per week, with the highest rate of weight gain during week 1 (1.25 ± 1.28 kg/week). Over 4 weeks, the total weight gain was 3.00 ± 1.92 kg. Nine patients (7.5%) developed mild hypophosphatemia, and none developed refeeding syndrome. Conclusions: Starting re-alimentation with 2000 kcal/d under close medical surveillance, severely malnourished youth with AN met the recommended weight gain targets between 0.5 and 1 kg/week according to current treatment guidelines, without anyone developing refeeding syndrome.

## 1. Introduction

Alimentation protocols in patients with anorexia nervosa (AN) have long been defined and restricted by the perceived risk of developing refeeding syndrome (RS) [1]. RS is commonly understood as a series of electrolyte derangements preceding organ dysfunction and rhabdomyolysis, seizures, delirium, coma and—in rare cases—death [2]. Studies have identified NG feeding, weight gain, electrolyte abnormalities and BMI on admission as potential indicators of developing refeeding syndrome in youth [3]. In particular, low phosphate levels have been shown to result in a drop in intracellular ATP levels, causing common complications of RS [4,5,6]. Falling serum phosphate levels have therefore been used as a risk marker of RS. However, data on incidence, mortality and safety indicators for this complication are controversial, and thus no clear consensus exists on the magnitude and ability to mitigate the risk of fulminant RS during the alimentation of malnourished patients with AN. In adolescents, a systematic review showed that the average incidence of refeeding hypophosphatemia, defined as serum phosphate below 0.9 mMol/L, was 14%, with a range of 0–48% [7].

To minimize the risk for RS, consensus-based re-alimentation recommendations for adolescents with AN start with a low initial energy intake, e.g., 1200 kcal/d, with increases of 200 kcal every other day [8,9]. However, these approaches often contribute to initial weight loss and delayed or slow weight gain during re-alimentation [10]. Therefore, these low-calorie approaches [11] are increasingly being criticized and replaced by higher-caloric protocols (HCR, starting calories ≥ 1400 kcal). One US study used a quasi-experimental study design to compare lower-caloric vs. higher-caloric re-alimentation approaches in 56 adolescents with AN (age = 16.2 ± 0.3 years, baseline percent median body mass index (%mBMI) = 79.2 ± 1.5%). These patients were hospitalized in the same medical institution for a mean inpatient stay of 14.9 ± 0.9 days between 2002 and 2008, where low refeeding standards were used, and 2008 and 2012, the period of HCR implementation. Groups were split at a median of 1200 kcal refeeding initiation (1764 ± 60 kcal/d vs. 1093 ± 28 kcal/d) [12]. The results indicated that HCR did not result in increased RS, merely increasing the tendency to receive phosphate supplementation (12 vs. 8 participants, *p* = 0.0273), and proved to result in more effective weight gain (0.46 ± 0.04 vs. 0.26 ± 0.03 %mBMI/day, *p* < 0.001). A second US study reviewed the charts of 310 consecutively hospitalized youth between the ages of 10 and 21 (mean = 16.1 ± 2.3 years; female = 88%, %mBMI = 78.5 ± 8.3) and compared a lower- vs. higher-calorie re-alimentation protocol. In this study, the lower-calorie group (*n* = 88) began at 1163 ± 107 kcal/d (range = 720–1320 kcal/d), whereas the HCR group started at 1557 ± 265 kcal/d (range = 1400–2800 kcal/d) [13]. The results indicated that the HCR group had a reduced inpatient treatment duration (13.0 ± 7.3 days vs. 16.6 ± 9.0 days; *p* < 0.0001), but without differences in the incidence of hypophosphatemia (38.90% vs. 25.8%, *p* = 0.49). Notably, hypophosphatemia was associated with greater malnutrition on admission (*p* = 0.004), but not with the caloric intake regimen (*p* = 0.14). Similar results were described in studies with a longer observation period from Sweden. In a 12-week study of 21 adolescents and young adults aged 16–24 with AN (mean age = 19.9 ± 2.4 years, BMI = 15.4 ± 0.94 kg/m^2^), daily caloric intake started at 3264 ± 196 kcal/d in week one and decreased gradually during treatment to 2622 ± 331 kcal/d, resulting in a weight gain of 9.8 kg (0.82 kg per week) without clinical side effects associated with RS [14]. Likewise, Koerner et al., using an HCR protocol targeting an initial energy intake of 2000 kcal/day, described in a retrospective chart review that, over four weeks, severely malnourished adults with AN (mean BMI = 11.5) treated in Germany gained a total of 4.2 kg (1.05 kg per week) and showed normalization of laboratory parameters with no case of RS [15]. The recently published short-term results of a multicenter randomized controlled trial (RCT) in the US by Garber et al. [16] compared 111 youth aged 12–24 years (mean = 16.4 ± 2.5 years, females = 91%) with AN and m%BMI ≥ 60% (explicitly “owing to concerns for medical fragility”). From day one of treatment, the HCR protocol (*n* = 60; beginning at 2000 kcal/day and increasing by 200 kcal daily), compared with a lower-caloric re-alimentation protocol (*n* = 51; beginning at 1400 kcal/day and increasing by 200 kcal every other day), showed a significantly faster restoration of medical stability (hazard ratio = 1.67, 95% confidence interval (CI) = 1.10–2.53; *p*  = 0.01) and a 4.0-day (95%CI = −6.1 to −1.9) shorter hospitalization for patients. Moreover, there was no difference in the frequency of adverse events and electrolyte imbalances between the two groups, and only 5% developed hypophosphatemia. The studies mentioned are summarised in Table 1. 

While these studies provide initial evidence for both the feasibility and increased efficacy of HCR protocols in both adults and adolescents, there is no clear consensus on the safety and generalizability of HCR in patients with AN, especially those with severe malnourishment. Moreover, alimentation strategies still differ widely around the globe, and data on meal-based HCR predominantly come from the US and during short-term hospital stays with limited follow-up time. Moreover, the published data are largely based on moderately malnourished patients, defined by AND/ASPEN (Academy of Nutrition and Dietetics and American Society for Parenteral and Enteral Nutrition) as >70% mBMI [17]. The aim of this study was therefore to assess the outcomes of an HCR protocol for adolescents with severe AN treated in a specialist eating disorder hospital in Germany. Outcomes were defined as weight development over a period of 4 weeks, as well as blood parameters indicating electrolyte shifts associated with RS.

## 2. Materials and Methods

### 2.1. Study Population

Included in this study were AN patients with BMI < 15 kg/m^2^ on admission, treated at the Schoen Clinic Roseneck in Germany between 01/2014 and 10/2019. Patient inclusion criteria were based on previous studies in adolescent and young adult AN patients [18,19]:Years of age, 12–20;Primary diagnosis of AN (any type);BMI < 15 kg/m^2^;Minimum inpatient treatment duration of 4 weeks (28 days);Availability of laboratory results at baseline and 28 days, plus at least two complete blood work results from the three weekly intervals between these dates.

Exclusion criteria were: pregnancy, diagnosis of bulimia nervosa, schizophrenia or psychosis.

All data, including laboratory data and information on the duration of the illness, comorbidities and medication received at baseline, were extracted from the medical files at the Schoen Clinic Roseneck, which were documented as part of routine clinical care. For patients not fulfilling the inclusion criteria, only the date of birth, hospitalization date, weight, height, BMI, BMI percentiles, %mBMI and BMI standardized (BMI-SDS) were extracted. For patients under the age of 18, the percentage median BMI (%mBMI) was calculated using the 50th BMI percentile for age and sex based on German reference data from Kromeyer-Hauschild [20]. To be comparable and consistent across the entire sample, %mBMI was also calculated for patients above the age of 18 [20].

Upon arrival, all patients signed a form in which they consented to the use of their clinical data for scientific purposes, and the chart review study received approval by the institutional ethics committee of the Ludwig Maximilian University in Munich (Project number 20–039 KB).

### 2.2. Setting, Re-Alimentation Protocol and Clinical Routine

The Schoen Clinic Roseneck is a German psychosomatic hospital specialized in the treatment of AN in both adolescents and adults. All patients begin the re-alimentation process with three meals per day, aimed at a mean initial caloric intake of 2000 kcal/day. This intake is divided into three main meals with a choice between vegetarian and non-vegetarian menus. During treatment, the aim is to draw the patients away from focusing on caloric intake, and meals are therefore described as a set unit. Micro- and macronutrients are not noted, but meals are prepared to contain sufficient protein, fat and carbohydrates. Daily caloric intake is increased individually for each patient according to achieved weight gain. In some refeeding protocols, caloric intake is related to the body weight of the patients, e.g., providing 30–40 kcal/kg of body weight per day at the start, and 70–100 kcal/kg of body weight per day at the end of treatment [21]. In contrast, the current refeeding protocol does not involve different energy intakes at the start of treatment according to individual body weight but provides 2000 kcal/d for all patients, with individual caloric increases in accordance with body weight development.

As per the HCR protocol, target weight gain was set between 700 and 1000 g/week, and patients were weighed regularly twice a week. In case of critical underweight (BMI < 13.5 kg/m²), weighing was increased to three times a week up to daily. The present study had no standardized protocol for weight increments and time intervals at which caloric intake was increased. Daily caloric intake was increased individually for each patient according to achieved weight gain. If the three main meals per day did not result in 700–1000 g/week of weight gain, an increase in the amount of food for the main meal (+50%), further snack meals and/or liquid drinks and, in rare cases, nasogastric or, in very severe cases, percutaneous endoscopic gastrostomy tube feeding (with the consent of patients and/or legal guardians) could be prescribed. The therapeutic approach is to specifically not communicate the caloric content of the food to the patients, but to describe increases in meal plans in terms of meal portions also used by healthy people. As an example, the patient would learn that after starting with the basic orientation plan on day one of treatment, if there is no sufficient weight gain within a week, the meal plan is increased by adding additional snack meals or increasing the portion size of the lunch meal by 50% while also paying attention to age, level of physical activity, bingeing, purging and tolerance of the food quantity. In order for the readers to judge the caloric content of this re-alimentation protocol, the hospital dieticians calculated the following: initially, all patients received 2200 kcal/d; increasing lunch by 50% in meal size has a caloric equivalent of 300 kcal, and snacks or liquid meals were added with a caloric equivalent of 300–400 kcal. A typical patient would thus, as an example, receive 2800–2900 kcal/d after 8 weeks of inpatient treatment, and at discharge after 4 months, there is high variation in energy needs, and meal plans can contain up to 3600 kcal/d.

All meals were accompanied and supervised by therapists; however, as patients progressed through their AN treatment, they were given more autonomy in eating and choosing their meals and portion sizes. Additionally, all patients took part in an intensive therapy program adapted to the individual needs of adolescents with AN, specifically addressing fears associated with the amount of food intake and weight gain. This treatment program included daily contacts and discussions with medical and psychological staff. Patients further received multimodal therapy based on cognitive behavioral therapy that included supervised meals and meal preparation classes, food intake protocols, body image exposure, sports and movement therapy adapted to weight and physical condition, individual and group psychotherapy sessions and specialized group therapy sessions (e.g., social skills, mindfulness, building healthy exercise habits). The positive effects of this treatment package have been shown in a one-year follow-up study published by Meule et al. [22]. Additionally, patients received regular blood tests and physical examinations to monitor the physiological response to the HCR protocol. This included blood tests on admission as well as a comprehensive medical exam, including ECG, full child psychiatric evaluation and fluid status, in addition to routine follow-ups depending on the test results. The interval at which laboratory values are assessed in this clinic is not specifically and prospectively defined in a standardized way but rather depends on the clinical status of the patient during treatment. In order to portray and examine the effect of HCR in this paper, only patients with blood values at baseline and day 28 and at least two of three further weekly blood values were included in this study.

On treatment initiation, breakfast consisted of two bread rolls, two packs of butter of 10 g or one pack of margarine of 20 g, eggs (optional), various condiments and a maximum of two cups of hot drinks and one glass of water or juice. Lunch usually entailed one optional serving of soup, a main meal with salad and dressing and dessert with one or two glasses of water (the second glass if the protocol-consistent amount of food was eaten). Dinner consisted of 3 slices of bread, 2 packs of butter of 10 g or 1 pack of margarine of 20 g, 1–2 slices of topping per slice of bread, 1 large cup of hot drink or the equivalent amount of water and no more than 2 small bowls of side dishes (raw vegetables/fruit). As energy intake was not objectively assessed, the individual macronutrient content of the meals was not known in this study. Hospital food in general, according to the dietician and hospital kitchen, contains 30% fat, 15–20% carbohydrates and 50–55% protein. The micronutrient composition of the hospital diet is not specifically defined.

Patients did not routinely receive prophylactic phosphate substitution. However, if clinically necessary (i.e., significant underweight below BMI 13 kg/m², neurological and cognitive deficits indicating the onset of refeeding syndrome or other values suggesting refeeding syndrome, such as CK elevation), clinical staff would measure the serum levels of phosphate and thiamine with substitution if necessary. Routine blood screening included electrolytes, white and red blood counts, liver enzymes, thyroid hormones, glucose, creatinine, creatine kinase, vitamin D and CRP on admission. If values were aberrant, controls were performed at least weekly.

### 2.3. Evaluation of Clinical Status during Intital Re-Alimentation

To assess the physical effects of the HCR process, patients received weekly laboratory controls. For this study, the following weekly (days 0, 7, 14, 21 and 28) parameters were defined to assess the effect of the HCR protocol on the physical well-being of patients with AN: full blood count, sodium, potassium, blood glucose level, calcium, phosphate, magnesium, chloride, creatinine, creatine kinase, y-GT, albumin and total protein. Additionally, GOT and GPT were used to assess potential liver cell damage, and leukocytes, hemoglobin and thrombocytes to assess bone marrow function.

The definition of refeeding syndrome used in this study is based on that of Rio et al. [23], in which all of the following criteria must be met to lead to a diagnosis of RS:Critically low serum electrolyte concentrations:
○Potassium < 2.5 mmol/L;○Phosphate < 0.32 mmol/L;○Magnesium < 0.50 mmol/L.
Peripheral edema or acute circulatory fluid overload.Disturbance of organ function, including respiratory failure, cardiac failure and pulmonary edema.

Consistent with previous guidelines [2], our study focused on serum phosphate concentrations as an indicator of impending RS.

### 2.4. Statistical Analysis

After characterizing the study population, a descriptive presentation of weight patterns as well as the trajectory of the above-mentioned physiological parameters was provided. Basic demographic characteristics between included and excluded patients were compared with tests for normal distribution, mean and range and tested for significance. The data were tested for normal distribution using the Shapiro–Wilk test. The appropriate parametric or nonparametric tests were then selected to examine the weekly changes in related outcomes in order to interpret these compared to reference values for normalization or derailment. Additionally, body weight change was also analyzed in a subgroup analysis comparing restrictive and active subtypes of AN (only 1 patient had the atypical subtype of AN, precluding a subgroup analysis). Data are presented as the mean ± SD (range) for normally distributed parameters, and as the median (25, 75 interquartile range (IQR)) for non-normally distributed parameters. To assess whether there was an association between body weight at admission and change in serum phosphate levels, Pearson’s correlation coefficient was calculated.

## 3. Results

### 3.1. Characterization of the Study Population Including Medical Details, Comorbidities and Medications

Between 2014 and 2019, 599 patients aged 12–19 years with a BMI of ≤15 kg/m^2^ and a primary diagnosis of AN were admitted to the Schoen Clinic Roseneck. Of these, 479 were excluded due to a lack of sufficient medical data. This was mainly due to a duration of stay of less than 28 days or a lack of laboratory values at baseline and 4 weeks as well as at two intervals in between. No serious clinical complications were observed among the excluded patients, but data are not reported due to the lack of weekly statistics. The remaining 120 patients included in the study were mostly female (*n* = 119; 99%) with a mean age of 17.3 ± 1.8 years; a total of 71 patients were <18 years old, and 49 patients were 18–19 years old. While there was no large difference in the average height of the included and excluded patients, the excluded patients had a significantly higher weight, BMI and BMI-SDS on admission than the included patients (Table 2).

The diagnoses, comorbidities and medications of the included patients are shown in Table 3. Altogether, 103 (85.8%) patients were diagnosed with restrictive-type AN (*n* = 103, 85.8 %), while 16 (13.3%) patients were diagnosed with active-type AN. Thirty-seven patients (30.8%) received phosphate supplementation, with a dosage of either 360 mg or 613 mg, for an average of 37 days ± 24 (range: 4–95). The different dosages were due to a lack of standard supplementation guidelines at the time of treatment. Three patients received nasogastric feeding (NG) tubes during the re-alimentation process. Information on the duration of NG tube feeding could not be retrieved in retrospect. On average, the three patients with NG tubes lost a total of 0.375 kg during the four-week observation period of this study. Two of these patients showed a total weight loss of −1.3 kg and −3.5 kg over 28 days. The third patient did not gain weight for the first 2 weeks but then showed an overall weight gain of 1.5kg over 28 days. No patient had RS, but seven patients (5.8%) had to be transferred to a pediatric psychiatric ward due to psychiatric complications (i.e., acute psychiatric emergency). Altogether, 24 patients (20.0%) discontinued treatment against medical advice after the 28-day observation period, and 1 patient (0.83%) had to be discharged due to a lack of compliance.

### 3.2. Changes in Body Weight

At baseline, the mean weight in kilograms was 35.6 ± 4.0 kg. The initial caloric intake per kg was therefore 57 kcal/kg ± 60.7. The mean BMI for adolescent patients was 13.2 ± 1.0 (range: 10.7–14.8) kg/m^2^. This is equivalent to a %mBMI of 64.1 ± 5.9 and BMI-SDS of −4.3 ± 1.1 (range: −6.5 to −2.6). Young adult patients (18 years and older) had a mean baseline BMI of 12.9 ± 1.2 (10.2–15.0) kg/m^2^. Body weight significantly differed across weeks (*p* < 0.001) and increased by 1.3 ± 1.3 kg in week 1, 0.6 ± 0.8 kg in week 2, 0.7 ± 0.8 kg in week 3 and 0.6 ± 0.9 kg in week 4. Between the initial hospitalization and day 28, patients experienced a mean weight gain of 3.0 kg, which is equivalent to 0.76 kg/week. This resulted in an increase of 1.1 ± 0.7 (−1–3–3.3) BMI points. The mean total weight gain was 10.0 kg ± 4.9 (range: −4.1–19.6) at an average duration of stay of 113 ± 48 (29–232) days (=3.76 months), leading to an overall significant increase in the mean BMI from 13.1 ± 1.1 (range: 10.2–15.1) kg/m^2^ at baseline to 16.7 ± 1.8 (range: 12.0–20.6) kg/m^2^ at discharge. Weekly changes in weight and BMI are shown in Figure 1a,b. The difference in weight gain between types of anorexia nervosa showed the following results: Patients with restrictive-type AN showed an increase in weight from 35.6 ± 4.1 on admission to 38.7± 4.4 by week 4. Patients with active-type AN showed an increase in weight from 35.9 ± 3.4 to 38.5 ± 2.9 by week 4. There was no significant difference between the weight changes of different AN subtypes (restrictive AN, 3.1 ± 1.9 vs. active AN, 2.6 ± 2.4, *p* = 0.637).

In week 1, 14 patients (11.7%) experienced weight gain between 0.1 and 0.5 kg, 18 patients (15.0%) gained between 0.51 and 1.0 kg, 43 patients (35.8%) gained between 1.01 and 2.0 kg, 21 patients (17.5%) gained between 2.01 and 4.0 kg and 4 patients (3.3%) gained more than 4.0 kg. The highest weight gain in week 1 was 5.7 kg. The consistent fall in the hematocrit of this patient showed that this can be traced back to hydration. Twenty patients (16.7%) showed weight loss or no weight gain during their first week of stay. Fifteen patients (12.5%) lost an average of −0.6 kg ± 0.5 (range: −1.8 to −0.1 kg) in weight in week 1. Of these, two continued to lose weight until week 4. Three patients failed to gain weight during the entire stay and were ultimately discharged due to a lack of compliance or were transferred to a closed psychiatric ward for intensified treatment and observation. Body weight and BMI trajectories are shown in Figure 1 and Table 4.

### 3.3. Changes in Blood Parameters

Changes in blood parameters over 4 weeks as well as the percentage of pathological values are shown in Table 5.

Trajectories of serum phosphate levels are shown in Table 6.

None of the patients had or developed RS. On average, phosphate levels increased from admission to week 4, and phosphate levels normalized in all patients (Table 5). However, to better understand the risk of refeeding hypophosphatemia in individual patients, it is important to also consider the development of hypophosphatemia over time. At baseline, twelve patients (10% of the study group) had low serum phosphate levels below the cut-off of 1.00 mmol/L. Reductions from normal to low phosphate levels occurred in seven patients (5.8% of the study population) in week 1, and in two patients (1.7% of the study group) in week 3, without any new development of low phosphate levels thereafter (Table 4). Only patients with restrictive-type AN showed pathological phosphate values on admission and throughout treatment. No patient with the active or purging AN subtype had pathological phosphate values.

Serum CK values decreased from baseline to week 4. Whilst 42 patients (35.0%) were above the cut-off of 123 U/L CK upon admission, this number was nearly halved by day 7, where only 23 patients (19.2%) had elevated CK levels, and dropped further to 18 patients (15.0%) by day 28. Mean pathological values decreased from 218.5 U/L at admission to 180.4 U/L by day 28, showing a clear decreasing trend in CK values in all patients. Whilst leukocytes and thrombocytes showed a significant increase over four weeks, Hb decreased continuously from 13.2 ± 1.5 (range: 8.0–16.0) upon hospitalization to 12.7 ± 1.15 (range: 8.7–15.0) by week 4. Similarly, hematocrit showed a drop from 39.9 (IQR: 36.6/42.1) to 38.7 ± 3.0 (range: 28.5–44.1). However, only one patient had a decrease in hematocrit to a critical level below 25% in week 1, which increased steadily into a normal range by week 4. A second patient experienced a decrease in hematocrit from 25.9 at admission to 20.3 at day 14 but then showed an increase to 28.5 by day 28. By week 4, no patient showed critical hematocrit levels below 25%. The weight gain resulted in a significant drop in elevated GOT and GPT levels from 30.4 (IQR: 24.5/39.4) and 29.6 (IQR: 20.7/53.5) at baseline to 23.7 (IQR: 18.9/28.2) and 25.6 (IQR: 19.9/35.7), respectively. The number of patients with elevated GOT levels dropped from 43 (36.1%) to 18 (15.0%) within 7 days and was down to 12 (10.0%) by day 28. While less patients showed complete GPT normalization within 28 days, the number of patients with elevated values also decreased from 52 (43.3%) to 42 (35.0%) in week 1 and further down to 31 (25.8%) within 4 weeks. Serum sodium values remained relatively stable, ranging from an average of 145 mmol/L at baseline to 142 mmol/L by day 28, with the number of patients with pathological values decreasing from six (5.0%) to two (1.7%). Overall, all laboratory values showed a significant normalization within four weeks.

Figure 1a–c show the development of weight, BMI and serum phosphate over the 28-day observation period, respectively.

### 3.4. Potential Correlates of Phosphate Levels

Serum phosphate and body weight on admission were not significantly associated (r = 0.09, *p* = 0.32, *n* = 118). Phosphate was also not significantly associated with the duration of illness (r= −0.76 *p* = 0.41, *n* = 120) and the number of previous hospitalizations due to AN (r = 0.10, *p* = 0.29, *n* = 111). However, there was a significant positive correlation (r = 0.27, *p* = 0.003, *n* = 118) between phosphate and BMI on admission (Figure 2) and BMI-SDS (r = 0.34, *p* < 0.01, *n* = 118).

## 4. Discussion

In this retrospective chart analysis, the outcomes of an HCR protocol were evaluated in 120 adolescent inpatients with AN with an admission BMI ≤ 15 kg/m^2^. This study found the following main results: (1) the HCR protocol resulted in an average weight gain of 760g per week during the first four weeks; (2) in this time, no cases of RS were observed, while nine patients (7.5%) developed hypophosphatemia, all phosphate levels were normal at 4 weeks and, overall, laboratory values normalized; (3) a low baseline BMI and hypophosphatemia were significantly associated, but individual variance around the regression line was high.

### 4.1. Energy Intake and Weight Gain

Few studies have described the effect of CR protocols on the rate of weight gain during hospitalization in severely malnourished adolescents with AN. Previous studies examining the weight trajectory during re-alimentation exhibited both higher [12,19] and lower [10] rates of weight gain than in the current study. A study conducted in the USA in 56 adolescent patients retrospectively compared weight gain over an average of 14.9 ± 9 days of inpatient admission between higher (1700–2800 kcal/day)- and lower (1000–2500 kcal/day)-caloric re-alimentation approaches [12]. In the high-caloric group, higher weight gain was observed (1.89 vs. 0.98 kg/week). A second study showed similar results, with patients achieving a weekly weight gain of 1.39 ± 1.49 kg over 15 days whilst receiving 1500 kcal/d on admission and increasing this up to 3600 kcal/d by day 14 [19]. However, compared to the population in the present study, malnutrition in both previous studies from the USA was markedly less pronounced, with a mean %mBMI of 79.2 ± 1.5 [12] and 79.4 ± 8.5 [19] compared to a %mBMI of only 64.1 ± 5.9 in the present study. Additionally, the observation time in this study was substantially longer and provides insight into longer-term weight development and changes in electrolytes. A third study [10] in the USA in 35 adolescent patients followed current American and German S3 Guidelines (starting at 1200 kcal and increasing this by 200 kcal every other day) and examined the weight trajectory along with caloric intake and clinical outcomes over a hospitalization time of 17 ± 6.4 days. Interestingly, 83% of the patients experienced initial weight loss, with peak weight loss at day 2, and only began to gain weight by day 8. Of note, the mean caloric intake was 1966 kcal per day by day 8, which is close to the initial caloric intake of 2000 kcal per day prescribed in our study. Nevertheless, twenty patients (16.7%) in the present study showed weight loss or no weight gain during their first week of stay, pointing towards a misconception in current guidelines, in that a weight gain of 500–1000 g/week can be achieved by starting with a low caloric intake (1200 kcal/d) in patients with AN. Our study demonstrates that an initial intake of 2000 kcal/d is necessary to achieve the weight target of 0.5–1.0 g/week and an immediate weight gain. This is an important finding which may help clinicians to provide adequate nutrition for patients with AN, and to avoid “underfeeding” and slow weight gain or even weight loss at the beginning of treatment.

### 4.2. High-Caloric Re-Alimentation and Occurrence of Hypophosphatemia or Refeeding Syndrome

To date, there is no consensus on the safety of HCR protocols in patients with AN with regard to the risk of hypophosphatemia and RS. While recent studies aimed to examine the metabolic and physical effects of HCR approaches in adolescents [10,12,24,25], the sample sizes were relatively small, and few studies have examined whether severely malnourished patients can tolerate such approaches [13,26]. In the present study on severely malnourished adolescents with AN, twelve (10.0%) patients presented with hypophosphatemia (<1.00 mmol/L) on admission. No patient had or developed RS, but nine patients (8.3%) with normal phosphate values on admission experienced a drop in phosphate levels below the reference range during treatment. Whilst the rate of hypophosphatemia on admission is relatively consistent throughout the literature (range: 7.9% [26] to 10.7% [19]), the reported incidence of hypophosphatemia (defined as values below 0.90 and 1.00 mmol/L) during treatment ranges from 15.8% [13] to 47.3% [19]. The occurrence of hypophosphatemia therefore varies greatly and is difficult to predict. While the incidence of refeeding hypophosphatemia may be affected by caloric intake and may be higher for HCR, the risk of RS is not [12,16]. Our study supports these findings and makes the important distinction that HCR was not associated with RS even in severely malnourished adolescents with AN. All patients with hypophosphatemia returned to the normal range without adverse events. One can therefore argue that HCR starting with 2000 kcal/d in an inpatient setting is a feasible alternative to current guidelines for severely malnourished adolescents with AN, in order to achieve faster and adequate weight gain. However, regular monitoring of serum electrolytes and physical well-being remains important to allow for early intervention and correction of any electrolyte imbalances which a significant proportion of AN patients present with at hospitalization.

### 4.3. High Variability in Supplementation Practices

Next to differences in caloric intake, the large heterogeneity in the reported numbers of hypophosphatemia could also be due to the varying practices of phosphate supplementation. It has been argued that supplementation of phosphate is safe [10,12,13,19] and may be more important in preventing the development of RH and RS than making use of conservative feeding approaches [24,27]. In our study, 31% of the patients received either 360 mg or 613 mg phosphate for an average of 37 days, whilst a study by Garber et al., in which 45% of the patients presented with hypophosphatemia, reported that 36% of the patients received a mean of 483g of phosphate supplementation [12]. Smith’s study [19] reported that 47.3% of the patients had hypophosphatemia, but 77.5% of the patients received phosphate supplementation, suggesting prophylactic use of supplements. Conversely, a study by Whitelaw et al. [28] supplemented serum phosphate only to treat hypophosphatemia. The use of phosphate supplementation is important in assessing the effect of HCR on serum phosphate. A recent narrative review on the current evidence of hospital protocols for patients with AN recommended the use of phosphate supplementation for patients with phosphate values < 1 mmol/L [29]. In this study, we found similar results with varying phosphate values and supplementation but argue that the benefits of prophylactic phosphate supplementation outweigh the potential side effects. Still, the lack of transparency in the nature and use of supplementation and its subsequent effects on patients limits the information on to which degree and in which form prophylactic or therapeutic usage of phosphate is effective and how it can be included in guidelines and recommendations. In line with this, the time intervals in which blood samples are drawn must be further defined, as this can also have an effect on treatment outcomes, and because results in the literature vary. A study by Murdeschwar et al. showed that within the first five days of daily testing, blood values remained within reference ranges and did not indicate impending refeeding syndrome, thereby arguing that daily blood testing for the first five days has limited use [30]. A study by Proulx-Cabana et al. recommended electrolyte testing within the first 3 days after admission and then twice weekly to continue screening for refeeding syndrome up to day 15 [29]. How quickly phosphate can drop into critically low values has not been defined in the literature, but in a study analyzing the few cases of reported RS, it was described that symptoms usually do not start within 72 h of admission but lasted at least 5 days.

### 4.4. Correlation between Hypophosphatemia and Degree of Malnutrition

Consistent with the previous literature [2,7,12,28,31], our study showed a positive relationship between the degree of malnutrition and hypophosphatemia on admission. However, while there may be a correlation between the two factors, one cannot say that hypophosphatemia is a phenomenon exhibited only at a lower BMI. In previous studies that examined the safety of HCR, it was shown that patients developed hypophosphatemia even with %mBMIs between 78 and 80 [7,12,13,19]. The degree of malnutrition could therefore be used as an indicator for pending hypophosphatemia but is not a definitive risk factor. Rather, consistent laboratory controls and physical examinations of the patient’s well-being can allow for early intervention in the case of electrolyte imbalances in the form of electrolyte substitution. Here, the possibility of prophylactic phosphate supplementation gains value, as it can be used in all patients to maintain normal serum phosphate values, thereby reducing the risk of developing RH and subsequently RS, while the adverse effects of phosphate supplements are not a concern [27].

### 4.5. Strengths and Limitations

The strength of this study lies in its relatively large sample size of 120 cases, application of a standardized HCR protocol and the comparatively longer observation time of 28 days compared to most prior studies. We were able to obtain detailed information from weekly electrolyte and other important blood work values as well as comprehensive weight trajectories, allowing for a comprehensive analysis of the effect of HCR. Furthermore, the youth included in this study were severely malnourished, with a BMI < 15, thereby extending prior work demonstrating the feasibility and safety of HCR for even more severely ill and medically unstable patients. This is also one of the first studies on high-caloric re-alimentation approaches in a German clinic in adolescents, demonstrating its feasibility in the European healthcare setting.

However, we also recognize the following limitations in our study. First, as this study was a retrospective chart analysis, this resulted in our reliance on accurate recording and allowing for limited follow-up data to examine the long-term benefits of HCR. Our study was a single-center study. Therefore, the generalizability to other settings may be limited. However, we were able to show that an HCR protocol did not result in refeeding syndrome and contributed to significant weight gain in accordance with guideline recommendations. This finding is in accordance with a US multicenter randomized clinical trial by Garber et al. [16], where an HCR protocol with an initial caloric intake of 2000 kcal/day and increases in calories of 200 kcal daily showed significantly faster restoration of medical stability (hazard ratio = 1.67, 95% confidence interval (CI) = 1.10–2.53; *p*  =  0.01) and a 4.0-day (95%CI = −6.1 to −1.9) shorter hospitalization for patients as compared to those who received 1400 kcal/day and an increase in calories of 200 kcal every other day. Further multicenter studies in different healthcare settings are needed to replicate these findings and to further examine the effects of HCR on different short- and long-term outcome dimensions. Moreover, the changes in calorie intake throughout the treatment were not objectively recorded to allow for a retrospective analysis of the increase in caloric intake and what metabolic and physical effects this has in severely malnourished youth. The caloric value per kg body weight throughout the treatment could therefore not be calculated. This calculation would be an interesting topic for further scientific research, as the effect of changes in calories in relation to the initial and increasing body weight could allow for weight-adapted HCR protocols. However, the objective documentation of the exact caloric intake for every patient could be difficult to implement.

Second, there was no objective way to ensure that each patient consumed a total of 2000 kcal/day. At the beginning of the treatment, all patient meals were accompanied by a therapist. Information on the daily caloric intake was therefore reliable. However, upon progression of the treatment, patients achieved more autonomy, and the information on caloric intake therefore became subject to the patient’s description. Nonetheless, assessing actual energy intake is difficult, and practical, clinical recommendations refer to the prescribed amount of energy leading to adequate weight gain. Third, this was a single-center study with no control group managed with a guideline-concordant lower-caloric re-alimentation treatment, which would have allowed for a direct comparison of outcomes in a German healthcare setting. Additionally, we did not collect data on reasons for patient exclusion from this study, which would have allowed for a more comprehensive comparison between included and excluded patient characteristics. Finally, we did not formally measure eating disorder psychopathology, (dis)stress, treatment satisfaction or potential compensatory behaviors (exercise, purging, laxative use) to assess non-physical adverse effects of the HCR protocol. Ideally, these factors should be assessed in future studies.

## 5. Conclusions

In summary, this study on the use of an HCR protocol in severely malnourished adolescents with AN demonstrated the positive effect of starting treatment with 2000 kcal/day to reach the guideline-concordant goal of a weight gain of 0.50–1.00 kg/day. This finding is important in reviewing current German and European recommendations of conservative re-alimentation approaches, as it is has become increasingly clear that HCR approaches are necessary to attain an adequate weight gain and prevent underfeeding and further weight loss. It is currently unknown whether adjusting the initial energy intake for individual body weights would improve weight outcomes compared with the current approach of providing 2000 kcal/d initially for all patients. Future studies are needed to compare the outcomes of different HCR re-alimentation protocols. Importantly, patients were able to physically tolerate this accelerated re-alimentation protocol well and exhibited no signs of critical deterioration or RS. For patients with AN, the initial 2000 kcal meal plan may seem like it includes large food portions at first and could potentially create emotional distress and/or gastric discomfort for the patients. However, the patients in the current study managed to follow the HCR meal plans and gain weight as required. Moreover, our data show that these meal plans were necessary to result in guideline-concordant weight gain. Less caloric intake may, in contrast, slow down weight gain or even lead to weight loss in already severely undernourished patients, prolong the period of starvation and potentially result in medical complications. Lower caloric intakes are known to prolong inpatient stays [1]. An interesting topic for further research would be to analyze weight gain, gastric discomfort and emotional distress in relation to portion sizes or different meal/re-alimentation protocols and how this would affect medium- to long-term weight recovery. The use of phosphate supplementation to prevent and treat refeeding hypophosphatemia also requires further research, as this precaution may be more important in preventing RS than the actual caloric intake. Large, multicenter RCTs should be conducted to confirm the safety, feasibility and efficacy of HCR in combination with specific prophylactic vs. therapeutic phosphate supplementation. Phosphate supplementation upon hypophosphatemia as well as the frequency of blood work analysis in relation to hypophosphatemia requires further research in order to reach a clear consensus on the utilization and prescription of phosphate in HCR protocols.

## Figures and Tables

**Figure 1 jcm-11-02585-f001:**
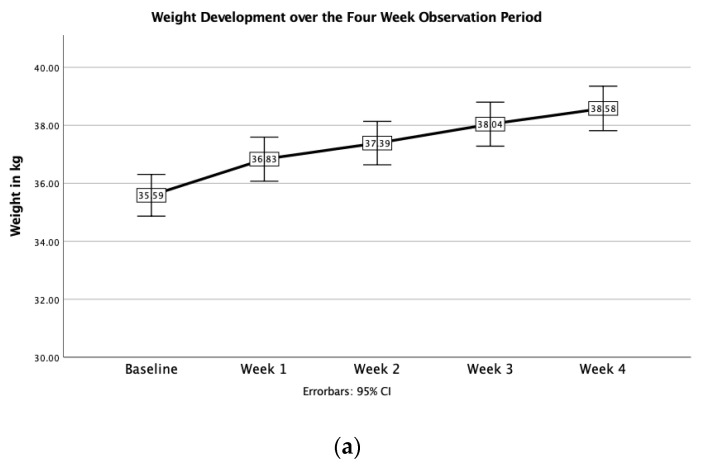
(**a**) Development of body weight over 28 days. (**b**) Development of BMI over 28 days. (**c**) Development of serum phosphate over 28 days.

**Figure 2 jcm-11-02585-f002:**
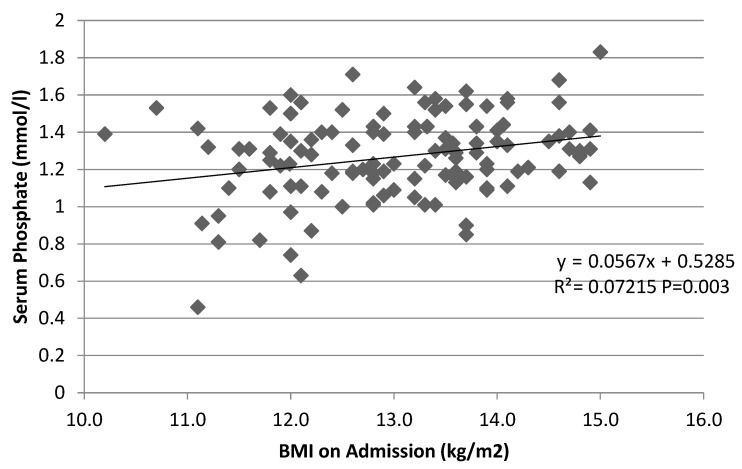
Relationship between serum phosphate and BMI on admission. While the correlation was significant, R^2^ was very low (r^2^ = 0.07), indicating only a weak relationship between the two factors. Patients with a lower BMI on admission tended to have lower serum phosphate values. However, the patient with the lowest BMI of 10.2 had normal serum phosphate values of 1.32 mmol/L, whereas a patient with a BMI of 13.7 presented with a phosphate value of 0.9 mmol/L. Moreover, the BMI of patients with hypophosphatemia in this study ranged from 11.1 to 13.7 kg/m^2^, indicating that low phosphate values are not necessarily exhibited only at lower BMI values.

**Table 1 jcm-11-02585-t001:** Previous studies regarding the effect of higher-calorie re-alimentation approaches on weight gain and other outcomes.

Study	Study Design	Follow-Up Duration (Weeks)	Patient Characteristics	Re-Alimentation Protocol	Weight Gain Outcome	Additional Outcomes
Garber et al., 2013 [12]	Quasi-experimental study design comparing lower- and higher-caloric re-alimentation protocols	14.9 (±0.9) days	56 adolescents with Anorexia Nervosa (AN) (age = 16.2 ± 0.3 years; baseline percent median body mass index (%mBMI) = 79.2% ± 1.5%)	Groups were split at a median of 1200 kcal refeeding initiation (1764 ± 60 kcal/d vs. 1093 ± 28 kcal/d)	(0.46 ± 0.04 vs. 0.26 ± 0.03% of median BMI (%mBMI)/day, *p* < 0.001).	No Refeeding Syndrome (RS), increased tendency to receive phosphate supplementation (12 vs. 8 participants, *p* = 0.0273)
Golden et al., 2013 [13]	Retrospective chart review comparing lower- vs. higher-calorie re-alimentation protocols		Youth between the ages of 10 and 21 (mean = 16.1 ± 2.3 years; female = 88%; %mBMI = 78.5 ± 8.3)	Lower-calorie group (*n* = 88) began at 1163 ± 107 kcal/d (range = 720–1320 kcal/d), Higher caloric realimentation (HCR) group started at 1557 ± 265 kcal/d (range = 1400–2800 kcal/d)		Reduced inpatient treatment duration (13.0 ± 7.3 days vs. 16.6 ± 9.0 days; *p* < 0.0001), no differences in the incidence of hypophosphatemia (38.90% vs. 25.8%, *p* = 0.49)
Pettersson et al., 2016 [14]	Observational study	12 weeks	Youth and young adults aged 16–24 with (AN) (mean age = 19.9 ± 2.4 years; Body Mass Index (BMI) = 15.4 ± 0.94 kg/m^2^)	3264 ± 196 kcal/d in week one, decreasing gradually during treatment to 2622 ± 331 kcal/d	9.8 kg weight gain in 12 weeks, (0.82 kg per week)	
Koerner et al., 2020 [15]	Retrospective chart review	4 weeks	Severely malnourished adults with AN (mean BMI = 11.5)	Initial energy intake of 2000 kcal/day	Mean total weight gain 4.2 kg in four weeks, (1.05 kg per week)	Normalization of laboratory parameters without any case of RS
Garber et al., 2020 [16]	Randomized controlled study	12 months	111 youth aged 12–24 years (mean = 16.4 ± 2.5 years; females = 91%) with AN and %mBMI ≥ 60% comparing higher-caloric and lower-caloric re-alimentation protocols	HCR: beginning at 2000 kcal/day and increasing by 200 kcal daily Lower-caloric re-alimentation: beginning at 1400 kcal/day and increasing by 200 kcal every other day		Significantly faster restoration of medical stability (hazard ratio = 1.67, 95% confidence interval (CI) = 1.10–2.53; *p* = 0 .01); shorter hospital stay: 4.0 days (95%CI = −6.1 to −1.9) for patients who received HCR; 5% developed hypophosphatemia

**Table 2 jcm-11-02585-t002:** Sample characteristics of included and excluded patients.

	Included Patients	Excluded Patients	*p*
*n*	120	479	
Age (years)	17.3 ± 1.8 (13.4–20.8)	16.6 ± 1.7 (12.9–20.9)	<0.001
Weight (kg)	35.6 ± 4.0 (25.2–44.0)	37.6 ± 4.0 (27.0–51.7)	<0.001
Height (cm)	165 ± 6 (151–180)	165 ± 7 (144–189)	0.784
Body Mass Index (BMI) (kg/m^2^)	13.1 ± 1.1 (10.2–15.0)	13.8 ± 0.9 (10.6–15.0)	<0.001
BMI percentile (*n* = 71 included; *n* = 382 excluded)	0.0 (0.0/0.0)	0.0 (0.0/0.1)	<0.001
(BMI-SDS) ^1^	−5.6 ± 1.87 (−6.3/−3.9)	−4.0 ± 1.3 (−4.6/−3.2)	<0.001
Duration of illness (months)	35 ± 23 (15/48)	Not applicable (NA)	NA
Length of stay	113 ± 48 (30–232)	107 ±55 (71/139)	0.247

^1^ BMI standard deviation scores.

**Table 3 jcm-11-02585-t003:** Clinical details including anorexia nervosa subtype, psychiatric comorbidities and medications at baseline for patients included in the data analysis.

Diagnosis	*n*	Percentage
Anorexia Nervosa		
Restrictive	103	85.8
Active	16	13.3
Atypical	1	0.8
Psychiatric Comorbidities		
Depressive disorders	66	55.0
Moderate depressive episode (F32.1)	40	33.3
Recurrent depressive disorder, current episode severe (F33.2)	8	6.7
Severe depressive episode (F32.1)	7	5.8
Recurrent depressive disorder, current episode moderate (F33.1)	6	5.0
Mild depressive episode (F32.0)	3	2.5
Recurrent depressive disorder, currently in remission (F33.4)	1	0.8
Other recurrent depressive disorders (F33.8)	1	0.8
Obsessive compulsive disorders	14	11.7
Mixed obsessive compulsive disorder (F42.2)	9	7.5
Predominantly compulsive disorder (F42.1)	4	3.3
Other obsessive compulsive disorders (F42.8)	1	0.8
Anxiety disorders	9	7.5
Social anxiety disorder (F40.1)	6	5.0
Specific, isolated phobia (F40.2)	3	2.5
Other psychiatric comorbidities	6	5.0
Post-traumatic stress disorder (F43.1)	4	3.3
Mixed anxiety and depressive disorder (F41.2)	1	0.8
Depersonalization-derealization syndrome (F48.1)	1	0.8
Medications at Baseline		
None	55	45.8
Psychotropic medications	32	26.7
Antidepressants	27	22.5
Antipsychotics	4	3.3
Both	1	0.8
Non-psychotropic medications	33	27.5

**Table 4 jcm-11-02585-t004:** Weight and Body Mass Index (BMI) development.

	Baseline	Day 7	Day 14	Day 21	Day 28
Weight (kg)	35.6 ± 4.0	36.8 ± 4.2	37.4 ± 4.2	38.0 ± 4.2	38.6 ± 4.2
	(25.2–44.0)	(27.0–45.5)	(28.0–46.7)	(28.5–49.6)	(29.6–50.6)
Body mass index (BMI) (kg/m^2^)	13.1 ± 1.1	13.5 ± 1.14	13.7 ± 1.1	14.0 ± 1.1	14.2 ± 1.2
	(10.2–15.0)	(11.0–16.3)	(11.1–16.7)	(11.3–17.2)	(11.1–17.5)
percentage of median body mass index (%mBMI)	62.1 ± 6.0	64.1 ± 6.5	65.0 ± 6.4	66.1 ± 6.7	66.9 ± 6.9
*p*		<0.001	<0.001	<0.001	<0.001

Normally distributed data are displayed as means ± SDs (range); non-normally distributed data are displayed as medians (25th–75th percentiles).

**Table 5 jcm-11-02585-t005:** Laboratory values on admission and by week four and percentage of pathological values.

	CK	Phosphate	Hematocrit	Sodium	GOT	GPT	Leukocytes	Hemoglobin	Thrombocytes
Cut-Off for Abnormality	>123 U/L	<1.00 mmol/L	<36%	<135 mmol/L	>35 U/L	>35 U/L	<3.50 g/L	<12 g/L	200 g/L
**At admission**	91.0	1.26 ± 0.26	39.9	145 ± 3	30.4	29.6	4.39	13.21 ± 1.47	242
	(52.5/139.5)	(0.46–1.83)	(36.6/42.1)	(131–151)	(24.5/39.4)	(20.7/53.5)	(3.47/5.68)	(8.00–16.00)	(211/292)
**Week 1**	72.0	1.21 ± 0.28	37.6 ± 4.6	142.66 ± 2.92	24.3	29.0	4.24	12.45 ± 1.62	246
	(510/114.00)	(0.49–1.56)	(23.3–44.8)	(135.0–150.0)	(19.9/31.1)	(20.9/44.0)	(3.54/5.37)	(7.80–14.90)	(201/288)
**Week 2**	69.0	1.29 ± 0.16	37.2 ± 4.2	142.2 ± 4.4	22.6	26.9	4.21	12.23 ± 1.49	256
	(49.0/94.0)	(0.90–1.84)	(20.3–44.1)	(131.0–170–0)	(19.0/28.4)	(19.4/37.7)	(3.51/5.62)	(6.40–14.50)	(221/309)
**Week 3**	67.0	1.31 ± 0.18	37.6 ± 3.8	141.9 ± 2.80	22.8	23.6	4.56 ± 1.56	12.28 1.40	278
	(47.0/93.0)	(0.68–1.91)	(24.1–44.0)	(128.0–147.0)	(18.2/27.0)	(20.5/35.9)	(2.15–8.47)	(7.30–14.70)	(231/313)
**Week 4**	66.0	1.33 ± 0.15	38.7 ± 3.0	142 ± 2	23.7	25.6	4.88	12.70 ± 1.15	278 ± 66
	(54.0/92.0)	(1.06–1.77)	(28.5–44.1)	(135–148)	(18.9/28.2)	(19.9/35.7)	(3.79/6.03)	(8.70–15.00)	(144–546)
Abnormal at admission	42/117 (35.9%)	12/118 (10.2%)	26/119 (21.8%)	6/120 (5.0%)	43/119 (36.1%)	52/119 (43.7%)	31/119 (26.0%)	24/119 (20.2%)	21/119 (17.6%)
Abnormal after 4 weeks	18/117 (15.4%)	0/117 (0%)	22/118 (18.6%)	2/120 (1.7 %)	12/119 (10.0%)	31/119 (26.0%)	21/118 (17.5%)	32/118 (27.1%)	12/118 (10.2%)
*p*-value	<0.0011	0.002	0.020	0.086	<0.001	<0.001	0.017	<0.001	<0.001

A *t*-test was used for normally distributed differences in blood values between two measurement points; the Wilcoxon test was used for non-normally distributed differences. Data are displayed as the mean ± standard deviation with the range in parentheses (if normally distributed), or the median with the 25th and 75th percentiles in parentheses (if not normally distributed). CK, creatine kinase; GOT, aspartate aminotransferase; GPT, alanine aminotransferase.

**Table 6 jcm-11-02585-t006:** Development of phosphate values including a distinction between normal and pathologically low values over 4 weeks.

	Normal–Normal (n)	Normal–Low (n)	Low–Normal (n)	Low–Low (n)
Baseline–Week 1	101	7	8	4
Week 1–Week 2	109	0	9	2
Week 2–Week 3	116	2	1	1
Week 3–Week 4	116	0	4	0

## Data Availability

Not applicable.

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
