# Peer review of "Outcomes of a Standardized, High-Caloric, Inpatient Re-Alimentation Treatment Protocol in 120 Severely Malnourished Adolescents with Anorexia Nervosa"

_jcm, 2022, doi:10.3390/jcm11092585_

Round 1

Reviewer 1 Report

It is a very good study, for a single-center study to determine benefit of higher caloric re-alimentation in adults or moderately malnourished adolescence. Starting re-alimentation with 2000kcl/d under close medical surveillance, severely  malnourished youth with AN met the recommended weight gain targets between 0.5 to 1 kg/week according to current treatment guidelines, without anyone developing refeeding syndrome. However, for a primary study, this paper can make differentiated between types of AN in adolescence. If this can combined with multiple center study, it will increase the valuable conclusion. 

Reviewer 2 Report

Dear Authors,

This is an interesting paper, that addresses a very important topic. To enhance the readability and wider context for the audience, I have some points below which I believe should be addressed to make the paper stronger:

Abstract:

I would suggest to add the range of BMI and the gender characteristic - number/% of female (line 22)

Introduction:

Authors should consider presenting the most important information from the discussed studies in tabular form. This will greatly improve readability.

MAterials and Methods:

In the case of pediatric patients, especially the younger ones, shouldn't the BMI value be related to the percentile charts? The interpretation of the BMI value in the case of children or younger adolescents is different from that used for adults. Literally: a 20-year-old's BMI 15 cannot be compared to a 12-year-old's BMI 15. Could Authors comment on this?

I don't fully understand the realimentation protocol. There is information that the initial caloric value was 2000 kcal / d, while the authors further write that the caloric value was dependent on the achieved weight gain (line 143). So, were the patients getting more than 2,000 kcal? what was the calorific value per kg of body weight? Did the patients eat all the meals they served? In the case of 3 meals and 2000 kcal, it was rather large portions which are a problem for patients with anorexia, especially with a low BMI.

The authors should provide more details about the diet: what was the share of energy from macronutrients, whether additional vitamin and mineral supplementation was used, whether "Nutridrink" type preparations were used, whether the nutritional value of the diet was adjusted to the individual needs of patients due to their age, gender and condition health / nutrition. Were the patients receiving 2000 kcal / d for the entire period of re-feeding? This information is missing and is essential for other researchers to follow the same patient management protocol.

Results:

Changes in body weight during therapy should be presented in the table for better readability.

I believe that taking into account the results of converting the caloric value of the diet to kg of body weight and showing the effectiveness of re-alimentation in such an approach could give interesting and more objective results.

Round 2

Reviewer 2 Report

Previous comments were adequately addressed in this version of the manuscript.